# Tumor-derived exosomal miR-1247-3p induces cancer-associated fibroblast activation to foster lung metastasis of liver cancer

Tian Fang[1], Hongwei Lv[1], Guishuai Lv[1,2], Ting Li[1], Changzheng Wang[1], Qin Han[1], Lexing Yu[1,2], Bo Su[3], Linna Guo[1,2], Shanna Huang[1,2], Dan Cao[1,2], Liang Tang[1,2], Shanhua Tang[1,2], Mengchao Wu[1,2], Wen Yang[1,2] & Hongyang Wang[1,2,4]

The communication between tumor-derived elements and stroma in the metastatic niche has a critical role in facilitating cancer metastasis. Yet, the mechanisms tumor cells use to control metastatic niche formation are not fully understood. Here we report that in the lung metastatic niche, high-metastatic hepatocellular carcinoma (HCC) cells exhibit a greater capacity to convert normal fibroblasts to cancer-associated fibroblasts (CAFs) than low-metastatic HCC cells. We show high-metastatic HCC cells secrete exosomal miR-1247-3p that directly targets B4GALT3, leading to activation of β1-integrin–NF-κB signaling in fibroblasts. Activated CAFs further promote cancer progression by secreting pro-inflammatory cytokines, including IL-6 and IL-8. Clinical data show high serum exosomal miR-1247-3p levels correlate with lung metastasis in HCC patients. These results demonstrate intercellular crosstalk between tumor cells and fibroblasts is mediated by tumor-derived exosomes that control lung metastasis of HCC, providing potential targets for prevention and treatment of cancer metastasis.

[1] International Co-operation Laboratory on Signal Transduction, Eastern Hepatobiliary Surgery Institute, Second Military Medical University, Shanghai, 200438, China. [2] National Center for Liver Cancer, Shanghai, 201805, China. [3] Central Laboratory, Shanghai Pulmonary Hospital, School of Medicine, Tongji University, Shanghai, 200433, China. [4] State Key Laboratory of Oncogenes and Related Genes, Shanghai Cancer Institute, Renji Hospital, Shanghai Jiaotong University, Shanghai, 200032, China. Tian Fang, Hongwei Lv and Guishuai Lv contributed equally to this work. Correspondence and requests for materials should be addressed to W.Y. (email: woodeasy66@hotmail.com) or to H.W. (email: hywangk@vip.sina.com)

Lung metastasis is the most frequent distant invasive progression and one of the main causes of cancer-related deaths in hepatocellular carcinoma (HCC)[1,2]. The process involves several steps driven by intercellular communications among various cells in the tumor microenvironment, including tumor cells and stromal cells[3,4]. Recently, therapeutic strategies that target tumor microenvironment components have become a compelling option in the fight against tumor metastasis[5,6]. As the most abundant cell type of tumor stroma, cancer-associated fibroblasts (CAFs), an activated sub-population of fibroblasts, have a key role in promoting tumor progression and metastasis[7–9]. Stemmed from different origins, CAFs are highly heterogeneous and expressed different specific markers for identification[10,11]. Among them, α-smooth muscle actin (α-SMA) is the most commonly used marker for CAFs[12]. Moreover, CAFs are believed to regulate the inflammatory microenvironment by expressing pro-inflammatory genes such as *IL-1β*, *IL-6*, *IL-8*, *TGF-β*, *CXCL12*, and *Collagen*[8,9,13–15]. The crosstalk between tumor cells and CAFs has been studied extensively[16–19]; however, the mechanisms underlying activation of fibroblasts by tumor cells remain unclear in liver cancer, even more obscure in lung metastasis from liver cancer.

Exosomes, measuring from 30 to 100 nm in diameter, are microvesicles formed in multivesicular bodies, which release exosomes into the extracellular milieu by fusing with cytomembrane[20,21]. Exosomes can be produced by various types of cells and serve as mediators in intercellular communications by transporting information cargo, such as proteins, lipids, and nucleic acids[22]. Specific proteins highly enriched in exosomes, such as Tsg101, CD63, Hsp70, CD9, and CD81, are usually used as markers to identify exosomes[23,24]. Numerous research reports have pointed out that exosomes mediate regulation of the tumor microenvironment to promote cancer metastasis and progression[25–28].

MicroRNAs (miRNAs) are small, non-coding RNAs, which function by inhibiting the target messenger RNAs translation. Recently, studies indicate that exosomes contain a high level of miRNAs and exosomal miRNAs have been shown to contribute to immunomodulation, chemoresistance, and metastasis in multiple tumor types[29–32]. However, the relevance between tumor-derived exosomal miRNAs and lung metastasis has not been elucidated clearly in liver cancer.

In our study, we set out to make a further clarification on the reasons leading to the different degree of lung metastasis from liver cancer. We identify the critical tumor-derived exosomal miRNA to convert fibroblasts to CAFs in lung pre-metastatic niche through exosomes microarray detection comparison between high-metastatic cancer cells and low-metastatic cancer cells. Moreover, CAFs promote a set of properties of tumor cells via increasing the secretion of pro-inflammatory cytokines. The bilateral interaction between primary tumor cells and stromal cells in distinct organs further illuminates a new mechanism of lung metastasis from liver cancer and offers new opportunities for potential therapeutic strategies targeting lung metastasis.

## Results

**Tumor-derived exosomes regulate fibroblasts activation.** In metastatic niche, CAFs have been demonstrated to actively participate in the tumor metastasis progression[8]. Similar results were observed in tissues of lung metastasis from HCC patients (Supplementary Fig. 1a) and in liver cancer cells in mice models (Supplementary Fig. 1b), according to the high expression of α-SMA, which is the most effective marker. Thus, it is very important to get a further understanding on CAFs activation caused by tumor cells. In light of the fact that exosomes secretion

is an important way for tumors to induce systemic changes, it rationally leads to the question of whether exosomes participate in the activation of fibroblasts[33]. To make it clear, four liver cancer cell lines with different migration and invasion abilities were chosen, among which CSQT-2 and HCC-LM3 were high-metastatic cancer cells, versus HepG2 and MHCC-97L (Supplementary Fig. 1c). First, we isolated and purified exosomes from conditioned media of tumor cells through the standard exosome isolation method of ultracentrifugation. The cup-shaped structure, size and number of the isolated exosomes were identified by electron microscopy and Nanosight particle tracking analysis (Fig. 1a, b). Intriguingly, we demonstrated that much more exosomes were secreted from high-metastatic cancer cells than low-metastatic ones through a quantified analysis of exosomes isolated from an equal number of cells (Fig. 1b). In addition, the detection of characteristic HSP70, CD63, TSG101, CD9, and CD81 further verified that the isolated particles were exosomes (Fig. 1c). MRC5, a kind of human embryonic lung fibroblasts, was chosen as normal fibroblasts (NFs) in our model. To identify the delivery of exosomes, we labeled the tumor-derived exosomes and fibroblasts with Dio (green) or Dil (red), respectively. After incubation, confocal imaging showed the presence of Dio spots in recipient fibroblasts, suggesting that labeled exosomes released by different tumor cells were delivered to fibroblasts (Fig. 1d).

To further evaluate the different abilities of educating fibroblasts among the exosomes derived from the four liver cancer cells in vitro, we divided them into the following groups: CSQT-2 versus HepG2 (with different origins) and HCC-LM3 versus MHCC-97L (with the same origin). Migration assays were performed and showed that more fibroblasts migrated in CSQT-2 or HCC-LM3 group, compared with the control group, respectively (Fig. 1e). Wound-healing assays further confirmed that high-metastatic cancer cell-derived exosomes could improve the fibroblasts migration ability remarkably (Supplementary Fig. 1d). More importantly, fibroblasts educated by exosomes from high-metastatic cancer cells expressed higher level of pro-inflammatory genes, such as *IL-1β*, *IL-6*, *IL-8*, *TGF-β*, CXCL12, Collagen type I (COL1A1), Collagen type III (COL3A1), and Collagen type IV (COL4A1), which have important roles in modulating the inflammation microenvironment and promoting carcinoma development (Fig. 1f). It is known that activated fibroblasts have enhanced matrix adhesions, resulting in increased contraction of collagen gel matrices[8,9]. We therefore assessed the effect of exosomes derived from HCC cells on fibroblast-mediated collagen contraction. The contraction abilities of fibroblasts were markedly enhanced after treatment with high-metastatic cancer cell-derived exosomes in comparison with low-metastatic cancer cell-derived exosomes (Fig. 1g). In addition, in vivo, intravenous exosomes from CSQT-2 or HCC-LM3 markedly contributed to the formation of lung metastasis induced by HCC cell line SMMC-7721, which was not seen in the HepG2 or MHCC-97L group (Fig. 1h). Enrichment of CAFs was observed in the lung metastasis induced by the high-metastatic cancer cells, as verified by the expression level of α-SMA detected via immunohistochemistry assay (Supplementary Fig. 1e). Altogether, the above results suggest that exosomes derived from high-metastatic cancer cells contribute to fibroblasts activation to foster lung metastasis of liver cancer.

**MiR-1247-3p in exosomes mediates fibroblasts activation.** We next explored how tumor-derived exosomes activate fibroblasts. MiRNAs encapsulated in exosomes are abundant and have an important role in cell-cell communication[34]. Therefore, we hypothesized that tumor-derived exosomal miRNAs mediate fibroblasts activation. To identify the specific miRNAs involved,

we conducted microarrays to generate miRNAs profiles of exosomes derived from the aforementioned four liver cancer cell types. Divided into two groups as before, the results were compared and shown as heatmaps in Fig. 2a. Twenty-one of the most

upregulated miRNAs (fold change > 10) in both two high-metastatic cancer cell-derived exosomes were subjected to validation (Fig. 2b). Only miR-1247-3p obviously promoted inflammatory genes (*IL-1β*, *IL-6*, and *IL-8*) expression in MRC5

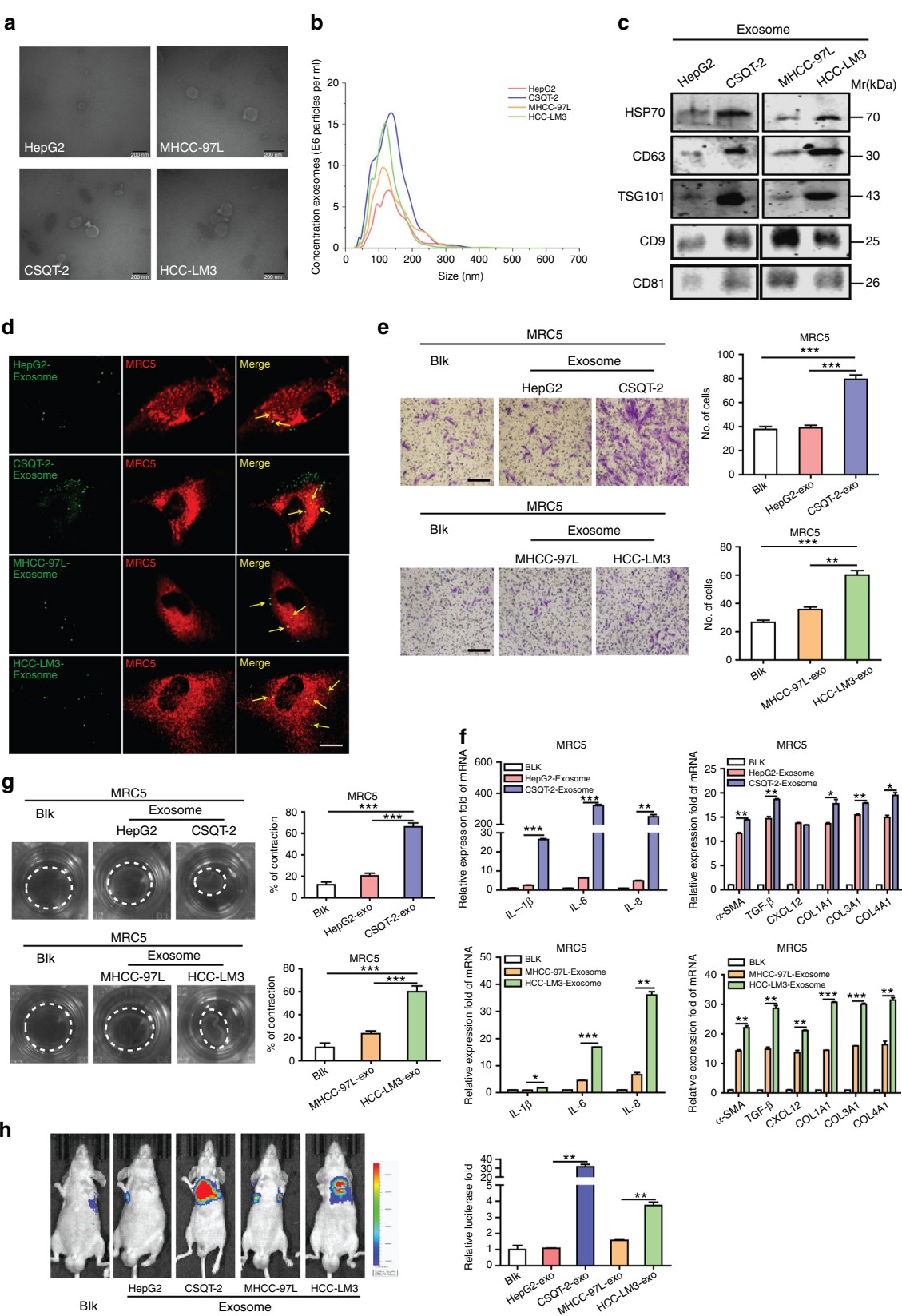

(Fig. 2c). Quantitative reverse-transcriptase PCR (qRT-PCR) analysis further confirmed the elevated expression of miR-1247-3p in both two high-metastatic cancer cells and high-metastatic cancer cell-derived exosomes (Supplementary Fig. 2a). In fact, in HCC cells, the expression of *IL-1β*, *IL-6*, and *IL-8* was also increased after miR-1247-3p treatment, suggesting the increased expression of these inflammatory genes may be a direct regulatory result of miR-1247-3p (Supplementary Fig. 2b). Furthermore, miR-1247-3p mimic also contributed to motility potential of fibroblasts (Fig. 2d and Supplementary Fig. 2c). To further investigate the role of miR-1247-3p, highly metastatic HCC cells were stably transfected with miR-1247-3p inhibitor (Supplementary Fig. 2d). As expected, the effect of miR-1247-3p on fibroblasts was abolished by its specific inhibitor (Fig. 2e, f and Supplementary Fig. 2e–g). Collectively, these findings reveal that tumor-derived exosomal miR-1247-3p mediates activation of fibroblasts.

**Exosomal miR-1247-3p directly targets B4GALT3 in fibroblasts**. For identification of the targets of exosomal miR-1247-3p in fibroblasts, two bioinformatics tools (miRDB and microT-CDS) were used to predict a set of common target genes. Among them, β-1,4-galactosyltransferases III (B4GALT3), a protein mediating glycosylation, was verified to be a direct target of miR-1247-3p and responsible for fibroblasts activation (Fig. 3a). First, we confirmed that B4GALT3 expression could be downregulated in MRC5 by miR-1247-3p or exosomes derived from high-metastatic cancer cells at both mRNA and protein levels (Fig. 3b and Supplementary Fig. 3a–d). Then the alignment between miR-1247-3p sequence and the full length of B4GALT3 sequence was determined to show that a B4GALT3 coding sequence was a potential target of miR-1247-3p (Fig. 3c). Subsequently, the wild-type and mutated miR-1247-3p-binding site were cloned into the luciferase vectors. It was obvious that luciferase activity decreased markedly in MRC5 co-transfected with the wild-type binding site vector in the presence of miR-1247-3p. However, cells containing the mutated binding site vector did not show such repression (Fig. 3d). Similarly, the luciferase activity remarkedly decreased only in MRC5 transfected with wild-type binding site vector of B4GALT3 after treatment with high-metastatic cancer cell-derived exosomes (Supplementary Fig. 3e, f). These results reveal that B4GALT3 is a direct target of miR-1247-3p in fibroblasts.

Furthermore, to determine the function of B4GALT3 in fibroblasts activation, we knocked down the B4GALT3 expression with small interfering RNAs (siRNAs) in fibroblasts and the effect was identified by qRT-PCR and immunoblotting analyses (Supplementary Fig. 3g, h). Next, migration assays and inflammatory gene expression analyses were performed. As shown in Fig. 3e, f, fibroblasts treated with siRNAs targeting B4GALT3 exhibited a promotion on motility and expression of inflammatory genes. Moreover, overexpression of full-length

B4GALT3 (Supplementary Fig. 3i, j) could neutralize the effect of miR-1247-3p on fibroblasts activation (Fig. 3g, h). Overall, these data suggest that B4GALT3 is a direct downstream target of miR-1247-3p and mediates fibroblasts activation.

**MiR-1247-3p activation of fibroblasts via B4GALT3-β1-integrin-NF-κB axis**. Results above showed that miR-1247-3p enhanced pro-inflammatory gene expression in MRC5, such as *IL-1β*, *IL-6*, and *IL-8*, which were well-known targets of nuclear factor-κB (NF-κB). Then we detected NF-κB signaling in our experiment models. As shown in Fig. 4a, c, conditioned medium (CM) of high-metastatic tumor cells promoted phosphorylated NF-κB expression, IκBα depression, and NF-κB signaling activation in MRC5, as compared with low-metastatic tumor cells. Similar results were obtained when MRC5 were treated with exosomes from indicated tumor cells (Fig. 4b, d). Furthermore, miR-1247-3p expression or B4GALT3 suppression also promoted NF-κB phosphorylation and IκBα depression (Fig. 4e), whereas B4GALT3 expression suppressed NF-κB phosphorylation (Fig. 4e). All these data indicate that miR-1247-3p activates fibroblasts by mediating NF-κB signaling activation.

As a member of B4GALTs family, B4GALT3 transfers galactose to *N*-acetylglucosamine to form *N*-acetyllactosamine in glycosylation of proteins[35]. Integrins are well-known cell surface receptors and consist of α- and β-subunits. It has been established that integrins promote nuclear translocation of NF-κB and NF-κB signaling activation to accelerate epidermal growth and migration[36]. Moreover, researches have provided evidence that B4GALT3 inhibits β1-integrin activation and stability by glycosylation modification, which increased *N*-acetyllactosamines content on β1-integrin[37]. Thus, the role of β1-integrin in the activation of NF-κB signaling induced by miR-1247-3p seems worthy of investigating. Immunoblotting analyses showed that β1-integrin expression was promoted by miR-1247-3p mimic or B4GALT3 knockdown; however, it was depressed by B4GALT3 expression (Fig. 4e). More importantly, the effect of miR-1247-3p on NF-κB phosphorylation was offset by β1-integrin knockdown (Supplementary Fig. 4a and Fig. 4f). In addition, it showed that B4GALT3 promoted degradation of β1-integrin when protein synthesis was blocked by cycloheximide (CHX) (Fig. 4g). Collectively, these results suggest that exosomal miR-1247-3p suppresses B4GALT3 expression to promote β1-integrin–NF-κB signaling in fibroblasts activation.

**Activated fibroblasts promote liver cancer progression**. CAFs are known to have an important role in tumor progression through the secretion of multiple pro-inflammatory cytokines and chemokines[8]. To determine whether fibroblasts educated by miR1247-3p contribute to the promotion of tumor characteristics, we conducted a set of experiments in vitro and in vivo. First, it was observed that overexpression of miR1247-3p enhanced the secretion of interleukin (IL)-6 and IL-8 in

**Fig. 1** Exosomes secreted from high-metastatic liver cancer cells regulate fibroblasts activation to foster lung metastasis. **a**, **b** Exosomes released by different cancer cells were detected by electron microscopy and Nanosight particle tracking analysis. Scale bar, 200 nm. **c** Immunoblotting assay of indicated proteins in exosomes from different cancer cells. **d** Confocal imaging showed the delivery of Dio-labeled exosomes (green) to Dil-labeled MRC5 (red). Yellow arrows represented delivered exosomes and representative images were presented. Scale bar, 25 μm. **e** Migration assays of MRC5 treated with equal quantities of exosomes derived from different liver cancer cells or blank control. Migrated cells were counted and representative images were shown. Scale bar, 150 μm. **f** Indicated gene expression of MRC5 treated with exosomes released by different liver cancer cells or blank control were detected by qRT-PCR analysis. **g** MRC5 contraction of collagen matrices. MRC5 treated with exosomes released by different liver cancer cells or blank control were assessed for their ability to contract collagen. Collagen contraction was quantified by the ImageJ software. **h** Representative images and quantitative analysis of lung metastasis of indicated mice treated with exosomes derived from different cancer cells were determined by luciferase-based bioluminescence imaging. Each experiment was performed three times independently and results are presented as mean ± s.d. Student's *t*-test was used to analyze the data. (**p* < 0.05; ***p* < 0.01; ****p* < 0.001)

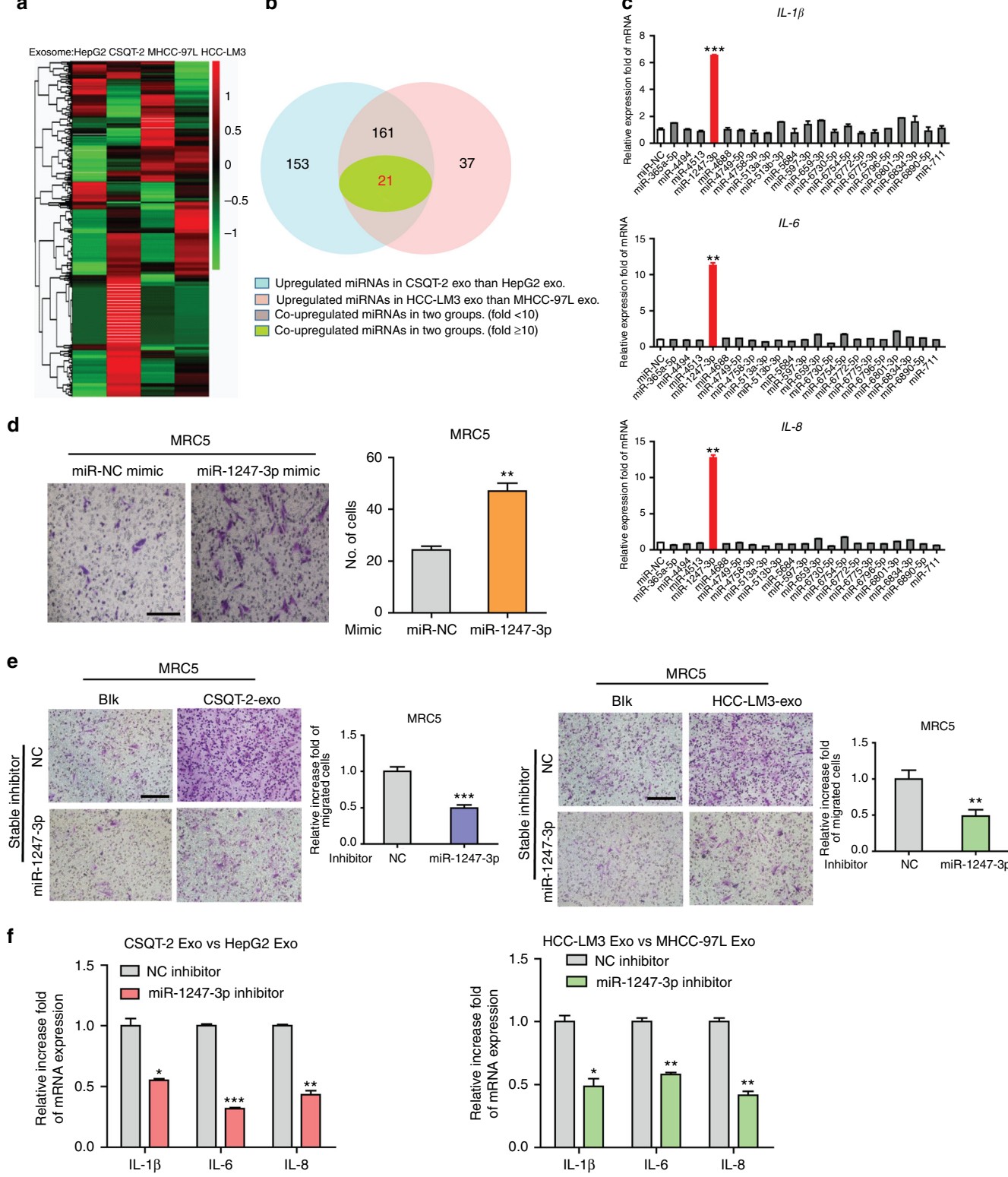

**Fig. 2** Exosomal miR-1247-3p is characteristically secreted by high-metastatic liver cancer cells and mediates fibroblasts activation. **a** Microarray analysis of exosomal miRNAs from different cancer cells were presented in a heatmap. **b** Overlapping results of upregulated miRNAs in indicated groups. **c** qRT-PCR analysis of pro-inflammatory genes expression of MRC5 transfected with indicated mimics. **d** Migration assay of MRC5 transfected miR-1247-mimic or normal control. Migrated cells were counted and representative images were shown. Scale bar, 150 μm. **e** Migration ability comparison of MRC5 treated with exosomes derived from CSQT-2 or HCC-LM3 with stably expressing miR-1247-3p inhibitor or negative control. Migrated cells were counted and representative images were shown. Scale bar, 150 μm. **f** qRT-PCR assay of indicated genes expression level of MRC5 treated with exosomes derived from HepG2 versus CSQT-2 or MHCC-97L versus HCC-LM3 in the presence of miR-1247-3p inhibitor or not. Experiments were performed at least in triplicate and results are shown as mean ± s.d. Student's t-test was used to analyze the data (*p < 0.05; **p < 0.01; ***p < 0.001)

fibroblasts (Fig. 5a). Then, tumor cells were educated by CM collected from fibroblasts pre-treated with miR1247-3p. As shown in Fig. 5b, c, e, f, h–j, after education, tumor cells exhibited enhancement of stemness genes expression, spheroid formation ability, motility, epithelial-mesenchymal transition (EMT)

process and resistance ability to sorafenib. Importantly, blocking IL-6 or IL-8 with neutralizing antibody partially reversed the increased spheroid formation ability, motility, and resistance ability to sorafenib of HCC cells after incubation with CM collected from fibroblasts pre-treated by miR1247-3p (Fig. 5d, g, i).

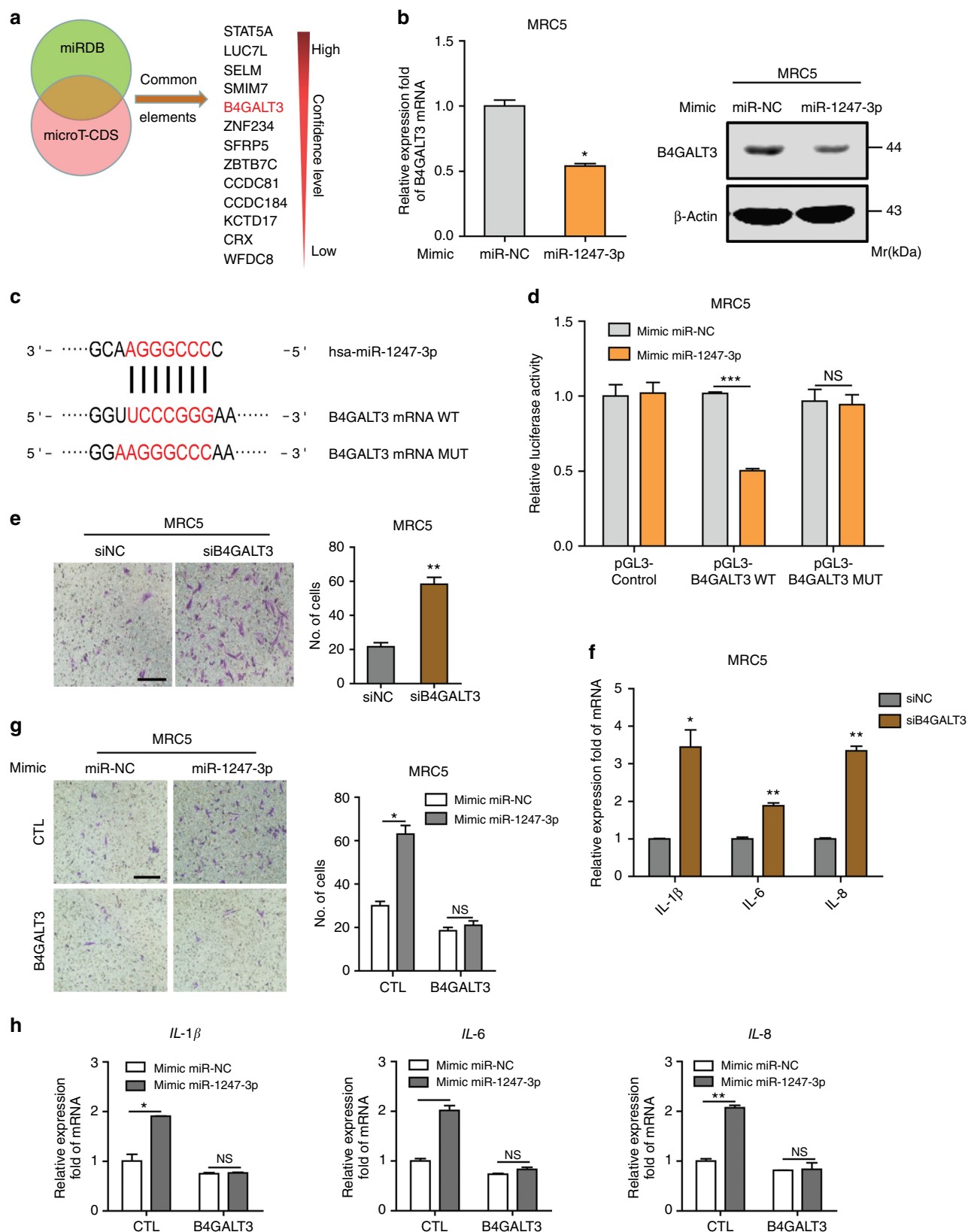

In addition, in vivo, CM of fibroblasts pre-treated with miR1247-3p, promoted tumorigenicity and proliferation of tumor cells (Fig. 5k). Consistently, exosomes derived from high-metastatic cancer cells also promoted the secretion of IL-6 and IL-8 in fibroblasts (Supplementary Fig. 5a). After treatment with CM collected from fibroblasts pre-treated by exosomes derived from high-metastatic cancer cell-derived exosomes, tumor cells showed increased spheroid formation ability, motility, and resistance ability to sorafenib (Supplementary Fig. 5b–d), and these phenomena were partially reversed by blocking IL-6 or IL-8 with neutralizing antibody (Supplementary Fig. 5e–g). These results suggest that fibroblasts educated by miR1247-3p promote tumor stemness, EMT, chemoresistance, and tumorigenicity in liver cancer.

To further study the function of CAFs in liver cancer, we isolated primary CAFs from invasive HCC patients with portal vein tumor thrombus and NFs isolated from normal liver tissues to serve as NFs. As expected, primary CAFs secreted higher level of IL-6 and IL-8 than NFs (Fig. 5l). Similar to fibroblasts educated by miR1247-3p, primary CAFs contributed to promoting tumor stemness, EMT, chemoresistance, and tumorigenicity (Supplementary Fig. 6a–h). More importantly, B4GALT3 expression levels decreased in primary CAFs, compared with NFs (Fig. 5m). In summary, similar interaction mechanism between tumor cells and fibroblasts through the transfer of miR-1247-3p may exist in intrahepatic metastasis niche of primary HCC.

**MiR-1247-3p correlates with lung metastasis.** To extend current knowledge to liver cancer patients, we first observed serum exosomes collected from HCC patients and healthy controls by electron microscopy and Nanosight particle tracking analysis (Fig. 6a, b). Parallel to findings before, a substantial increase of serum exosomes excretion was seen in HCC patients relative to healthy controls[38,39]. Then, we investigated exosomal miR-1247-3p expression in different serum samples (25 healthy controls, 90 HCC patients without lung metastasis, and 20 HCC patients suffering lung metastasis). As shown in Fig. 6c, elevated serum exosomal miR-1247-3p expression was detected in HCC patients, compared with healthy controls. More importantly, HCC patients suffering lung metastasis showed obviously higher levels of miR-1247-3p expression in serum exosomes than in patients without lung metastasis. To further determine the correlation of miR-1247-3p with clinicopathological features, in situ hybridization on 85 primary human HCC tissue samples was performed and confirmed that miR-1247-3p was differential expressed in liver cancer, similar to the results in serum exosomes (Fig. 6d). The HCC tissue samples were divided into two groups (low and high) according to the scores of miR-1247-3p expression level. As shown in Fig. 6e, high miR-1247-3p expression was well predicted for poor overall survival and poor disease-free survival. In addition, it was detected that high miR-1247-3p expression was correlated with increased alpha-fetoprotein (AFP) level, liver cirrhosis, tumor thrombus, and distant metastasis (Supplementary Fig. 7a, b and Supplementary Table 1). Taken together, these data show that miR-1247-3p is differentially expressed in liver cancer and high miR-1247-3p expression level in HCC tissues

predicted poor outcome. Moreover, high serum exosomal miR-1247-3p expression is correlated with lung metastasis of liver cancer. To further confirm the localization of miR-1247-3p expression in HCC tissue and lung metastatic tissue sample, in situ hybridization of miR-1247-3p a in combination with immunohistochemistry (IHC) staining of HCC markers (AFP) and fibroblast markers (α-SMA) were performed on serial sections of human HCC tissues and lung metastatic tissues. As shown in Fig. 6f, miR-1247-3p signals were present in both tumor cells and some fibroblasts around the tumor.

In sum, our results show that, tumor-derived exosomal miR-1247-3p converts fibroblasts to CAFs via downregulating B4GALT3, to activate β1-integrin–NF-κB signaling pathway to promote lung metastasis of liver cancer. Activated fibroblasts further promotes stemness, EMT, chemoresistance, and tumorigenicity of liver cancer cells by IL-6 and IL-8 secretion (Fig. 6g). Our findings also indicate that tumor-derived exosomal miR-1247-3p has an important role in intercellular communication for fostering an inflammatory microenvironment and promoting lung metastasis.

## Discussion

Tumor microenvironment, a dynamic system orchestrated by intercellular communications, is responsible for tumor progression and metastasis[40]. Therefore, it necessitates the study of the interaction between tumor and stroma mediated by exosomes. In our study, we first analyzed the different exosomal miRNA profiles between high-metastatic cancer cells and low-metastatic cancer cells. Then, we identified that miR-1247-3p was directly transferred from tumor cells to fibroblasts in lung pre-metastasis niche via exosomes and converts fibroblasts to CAFs by decreasing its target B4GALT3 expression to activate β1-integrin–NF-κB signaling pathway. In addition, CAFs promote tumor development by secreting IL-6 and IL-8. The crosstalk between tumor cells and fibroblasts further elucidates the molecular mechanism of lung metastasis from liver cancer, and explains why diverse liver cancer types show different abilities to induce lung metastasis. Furthermore, it is indicated that high miR-1247-3p expression in serum exosomes is correlated with lung metastasis of liver cancer, which holds important implications for efficient prevention and therapeutic strategies.

Previous studies show that miR-1247 is expressed aberrantly and has different roles in several cancers. For example, miR-1247 is downregulated and tends to act as a tumor suppressor miRNA in osteosarcoma. It is also determined that miR-1247 inhibits cell proliferation by targeting neuropilins in pancreatic cancer. However, castration-resistant prostate cancer shows high expression of miR-1247[41–43]. Our data demonstrate that tumor-derived exosomal miR-1247-3p converts fibroblasts to CAFs in lung pre-metastatic niche. Moreover, high expression of miR-1247-3p in serum exosomes has a positive correlation with lung metastasis from liver cancer. High miR-1247-3p expression in HCC predicts a poor outcome. Future work will be required to study the role of miR-1247-3p in liver cancer.

The integrins mediate cell adhesion and signals transmitting in multiple types of cells and have critical roles in cell proliferation

---

**Fig. 3** B4GALT3 is a direct downstream target of miR-1247-3p in fibroblasts activation. **a** Target gene prediction of miR-1247-3p with two bioinformatics tools. **b** qRT-PCR and immunoblotting assays of B4GALT3 expression in MRC5 treated with miR-1247-3p mimic or normal control. **c** The wild-type and a mutated type of binding site between miR-1247-3p and B4GALT3. **d** Relative luciferase activity of MRC5 in the presence of indicated treatments. **e, f** Migration assay and qRT-PCR analysis of MRC5 transfected with siRNAs targeting B4GALT3 or control. Scale bar, 150 μm. **g** MiR-1247-3p effect on migration ability of MRC5 in the presence of B4GALT3 or not. Migrated cells were counted and representative images were shown. Scale bar, 150 μm. **h** qRT-PCR analysis of pro-inflammatory genes expression in MRC5 with indicated treatments. Experiments were performed at least in triplicate and results are shown as mean ± s.d. Student's t-test was used to analyze the data (NS, not significant; *p < 0.05; **p < 0.01; ***p < 0.001)

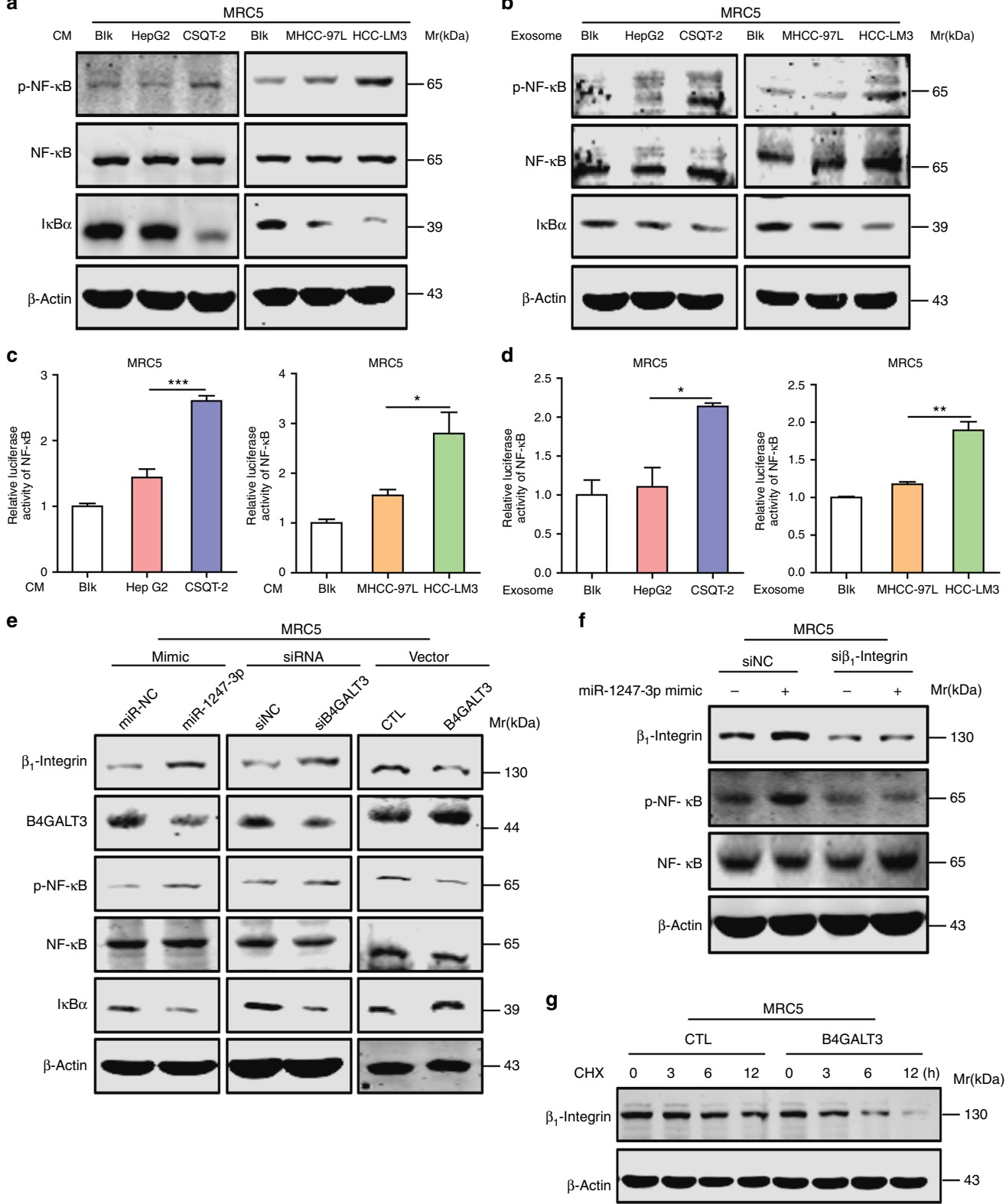

**Fig. 4** Exosomal miR-1247-3p activates fibroblasts via B4GALT3–β1-integrin–NF-κB signaling axis. **a**, **b** Immunoblotting assays of indicated proteins in MRC5 treated with CM or exosomes from different tumor cells. **c**, **d** Relative luciferase activity of NF-κB in MRC5 in the presence of CM or exosomes from different tumor cells. All data are shown as mean ± s.d. Student's *t*-test was used to analyze the data. (*$p < 0.05$; **$p < 0.01$; ***$p < 0.001$). **e** Immunoblotting assays of indicated proteins in MRC5 with indicated treatments. **f** MiR-1247-3p effect on indicated proteins expression in MRC5 with or without suppression of β1-integrin expression. **g** Degradation assay of β1-integrin in MRC5 transfected with B4GALT3 or not

and differentiation[36]. Recent studies have provided further understanding of integrins in cancer. For example, integrin β4 signaling has been shown to contribute to tumor angiogenesis[44]. Moreover, integrin patterns on tumor exosomes determine organotropic metastasis[45]. Herein, our results show that β1-integrin is glycosylated by B4GALT3 to alter self-stability and its enhanced expression mediates NF-κB signaling activation induced by miR-1247-3p in fibroblasts. As the combination

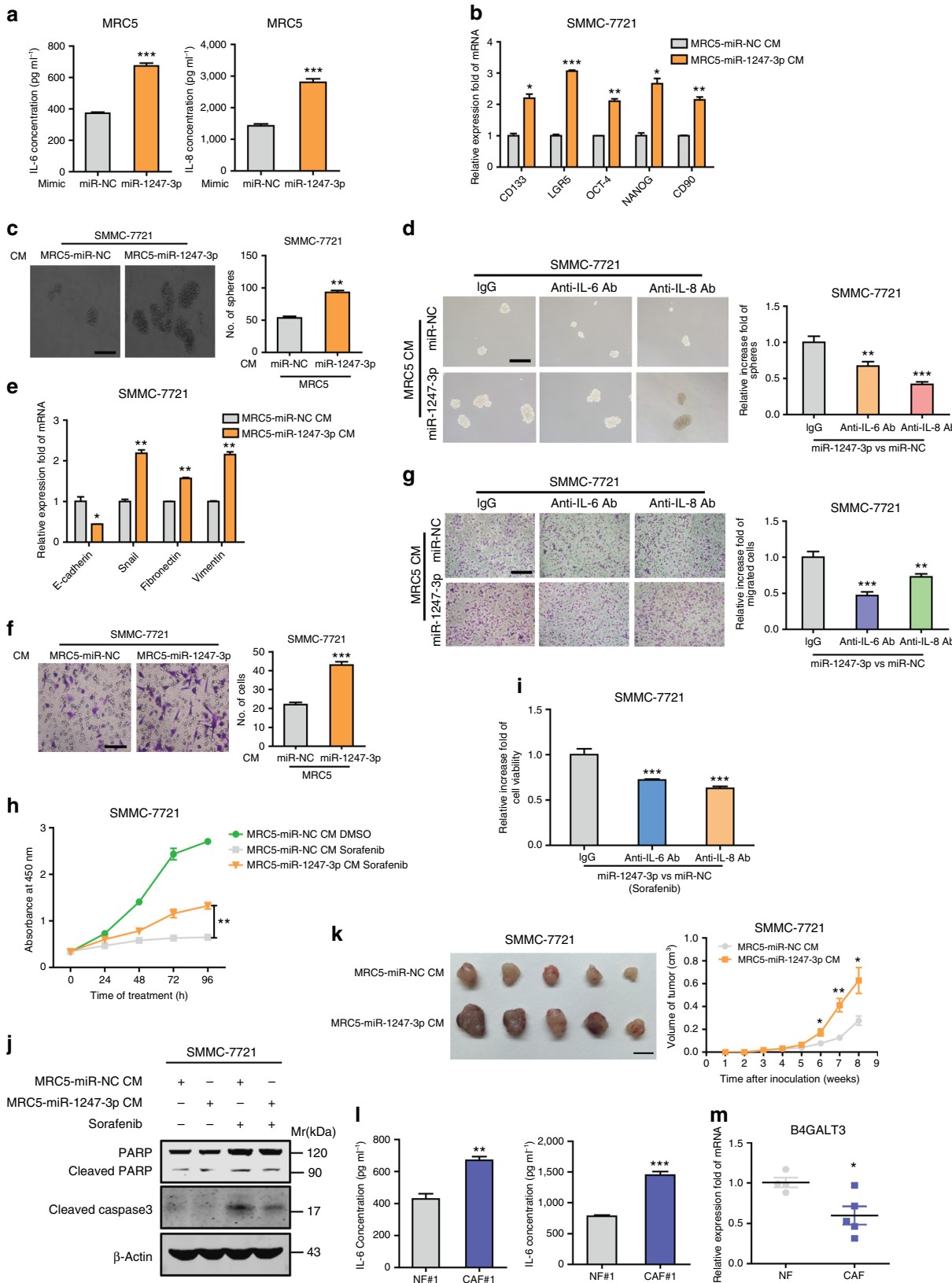

patterns of integrin subunits change diversely, the role of integrins in cancer remains to be thoroughly studied. Even so, treatments targeting integrins provide meaningful opportunities for tumor therapeutic strategies.

In conclusion, our results indicate that tumor-derived exosomal miR-1247-3p converts fibroblasts to CAFs by decreasing B4GALT3, to activate β1-integrin–NF-κB signaling pathway in lung pre-metastatic niche from liver cancer. In addition, CAFs exhibit increasing secretion of IL-6 and IL-8, promoting stemness, EMT, chemoresistance, and tumorigenicity of tumor cells. More importantly, high expression of miR-1247-3p in serum exosomes shows a positive correlation with lung metastasis in HCC patients. Our study elucidates a new molecular mechanism underlying the crosstalk between tumor cells and fibroblasts to promote lung metastasis, which contributes to efficient prevention and therapeutic strategies for liver cancer.

## Methods

**Specimens and primary cells**. Human serum specimens were collected from healthy donors and HCC patients before resection in the Eastern Hepatobiliary Surgery Hospital in Shanghai, China. A tissue microarray composed of HCC samples from 85 patients was also obtained from the Eastern Hepatobiliary Surgery Hospital. The clinicopathological features of 85 patients are displayed in Supplementary Table 1. The lung metastasis tissues from HCC were obtained from HCC patients in Shanghai Pulmonary Hospital. The human primary CAFs or NFs were isolated from fresh tumor tissues of HCC or normal liver tissues with the tumor dissociation kit (Miltenyi Biotec) and cultured in Dulbecco's modified Eagle's medium/F12 (DMEM/F12) (Gibco, Carlsbad, CA, USA), supplemented with 10% fetal bovine serum (FBS) (Gibco). All procedures were conducted with the approval of the Ethical Committee of the Second Military Medical University and Shanghai Pulmonary Hospital. Patient consent was obtained before the start of the study.

**Cell culture**. The human liver cancer cell lines HepG2, CSQT-2, MHCC-97L, HCC-LM3, and SMMC-7721 were purchased from Cell Bank of Type Culture Collection of the Chinese Academy of Sciences (Shanghai Institute of Cell Biology) and cultured in DMEM (Gibco) supplemented with 10% FBS (Gibco). The fibroblast cell line MRC5 was purchased from Sciencell Company (USA) and cultured in Eagle's minimum essential medium (Wisent, China) supplemented with 10% FBS (Gibco). All cell lines were cultured in a humidified incubator containing 5% $CO_2$ at 37 °C. Cell lines were authenticated by short tandem repeats (STR) profiling and confirmed to be mycoplasma negative.

**Reagents and antibodies**. Antibodies for TSG101 (ab125011, 1 : 1,000), HSP70 (ab2787, 1 : 1,000), α-SMA (ab32575, 1 : 100), Rab27A (ab108983, 1 : 1,000), β1-integrin (ab52971, 1 : 1,000), CD9 (ab92726, 1 : 1,000), and CD81 (ab79559, 1 : 1,000) were purchased from Abcam (Cambridge, MA, USA). Antibodies for phosphotylated NF-κB (S536) (3033, 1 : 1,000), total NF-κB (8242, 1 : 1,000), IκBα (4812, 1 : 1,000), poly ADP-ribose polymerase (PARP) (9532, 1 : 1,000), and Cleaved Caspase3 (9664, 1 : 1,000) were purchased from Cell Signaling Technology (Beverly, MA, USA). Antibodies for CD63 (A5271, 1 : 1,000) and B4GALT3 (A11939, 1 : 1,000) were purchased from ABclonal Technology (China). Antibody for β-actin (sc-47778, 1 : 1,000) was purchased from Santa Cruz Biotechnology (Santa Cruz, CA) and served as reference protein. Anti-IL-6 and anti-IL-8-neutralizing antibody were purchased from R&D Systems (Minneapolis, USA).

CHX was purchased from Sigma-Aldrich (St. Louis, MO, USA). Sorafenib was purchased from TargetMol (USA).

**Western blotting**. Whole-cell protein extracts were homogenized in lysis buffer and centrifuged at 12,000 r.p.m. for 15 min. Bicinchoninic acid (BCA) assay was performed to measure the protein concentrations. After immunoblotting, the proteins were transferred to nitrocellulose filter incubated with specific antibodies subsequently. The immunocomplexes were incubated with the fluorescein-conjugated secondary antibody, and then detected by an Odyssey fluorescence scanner (Li-Cor, Lincoln, NE). To detect indicated proteins, specific primary antibodies were used as follows. The uncropped scans of western blots from the main figures are displayed in Supplementary Fig. 8.

**RNA interference and plasmids**. siRNAs (siNC, siRNA targeting RAB27A, B4GALT3, or β1-integrin) and mimics of indicated miRNAs were conducted by Biotend Company (Shanghai, China). The sequences of siRNAs and miRNA mimic referred above were listed in Supplementary Table 2 and Supplementary Table 3. Lentivirus vectors containing miR-1247-3p inhibitor and control were constructed and generated by Genechem Inc. (Shanghai, China). The lentiviral transfected cells were selected with puromycin (Clontech, USA). The plasmid vector containing B4GALT3 and empty vector were conducted by Obio Technology (Shanghai, China).

**Animal studies**. For examining the roles of exosomes in lung metastasis models, $1 \times 10^6$ luciferase-labeled SMMC-7721 cells were intravenously injected into male nude mice through the tail vein (Chinese Science Academy, Shanghai, China). Subsequently, mice were divided into groups randomly and intravenously injected with equal number exosomes from different tumor cells twice a week for a month. After another month, lung metastasis was measured and quantified by ex vivo bioluminescent imaging using IVIS Lumina series III (PerkinElmer, USA).

For xenograft assays, $1 \times 10^6$ educated SMMC-7721 or control cells were injected subcutaneously into the right side of each male nude mouse (Chinese Science Academy). The sizes of tumors (length × width$^2$ × 0.5) were measured at the indicated time points and tumors were obtained after 4 weeks after injection. All animal experiments were approved by the University Committee on Use and Care of Animals of Second Military Medical University.

**Collagen contraction assays**. A total of $2 \times 10^5$ MRC5 were suspended in 100 μl DMEM. Then the cell suspension was mixed with 100 μl of collagen mix containing 68.75 μl DMEM, 0.72 μl 1 N NaOH, and 31.25 μl Type 1 Rat Tail Collagen (Corning), and added to 1 well of 24-well plates and allowed to solidify for 45 min at 37 °C. After incubation with media containing tumor-derived exosomes, the gels were photographed at different time points. ImageJ software was employed to measure gel area and evaluate contraction.

**Sphere formation and cell viability assay**. For sphere formation, indicated cells were plated on 6-well plates in a density of 3,000 per well and cultured in DMEM supplemented with 10% FBS. After observation under microscope, the number of spheroids was counted and representative images were captured. For cell viability assay, Cell Counting Kit 8 (CCK-8) assay (Dojindo Laboratories, Kumamoto, Japan) was used to assess cell viability according to manufacturer's methods.

**Migration assay and wound-healing assay**. For migration assay, $5 \times 10^4$ MRC5 were plated in 24-well transwell plates with inserts (Corning). The medium in inserts was free of FBS, whereas the medium out was supplemented with 10% FBS. For detecting exosome function, equal quantities of tumor-derived exosomes were added into the inserts. After 24 h, the cell inserts were fixed and stained according

**Fig. 5** Activated fibroblasts by miR-1247-3p promote liver cancer progression. **a** IL-6 and IL-8 secretion from MRC5 expressed miR-1247-3p or control were detected by ELISA assays. **b** qRT-PCR analysis of stemness-associated genes expression in SMMC-7721 with indicated treatments. **c** Spheroid formation ability of SMMC-7721 treated with indicated CM. Representative images were shown and spheroid were counted. Scale bar, 150 μm. **d** Relative spheroid formation ability of SMMC-7721 treated with indicated CM containing anti-IL-6/anti-IL-8 antibody or IgG control antibody. Representative images were shown. Scale bar, 150 μm. **e** qRT-PCR analysis of EMT-associated genes expression in SMMC-7721 with indicated treatments. **f** Migration assay of SMMC-7721 treated with indicated CM. Representative images were shown and migrated cells were counted. Scale bar, 150 μm. **g** Relative migration ability of SMMC-7721 treated with indicated CM containing anti-IL-6/anti-IL-8 antibody or IgG control antibody. Representative images were shown. Scale bar, 150 μm. **h** CCK8 assay of SMMC-7721 treated with indicated CM containing anti-IL-6/anti-IL-8 antibody or IgG control antibody in the presence of sorafenib. **i** Relative cell viabilities of SMMC-7721 treated with indicated CM containing anti-IL-6/anti-IL-8 antibody or IgG control antibody in presence of sorafenib. **j** Western blotting assay of indicated proteins in SMMC-7721 with indicated treatments. **k** Xenograft assays of SMMC-7721 with indicated treatments were performed on nude mice. Representative tumors (left) and tumors growth curves (right) were shown. **l** ELISA assay of IL-6 and IL-8 secretion from NF#1 or CAF#1 isolated from normal or primary HCC tissues. **m** B4GALT3 expression level was determined in isolated NFs and CAFs by qRT-PCR analysis. Each experiment was performed in triplicate and data are presented as mean ± s.d. Student's t-test was used to analyze the data (*$p < 0.05$; **$p < 0.01$; ***$p < 0.001$)

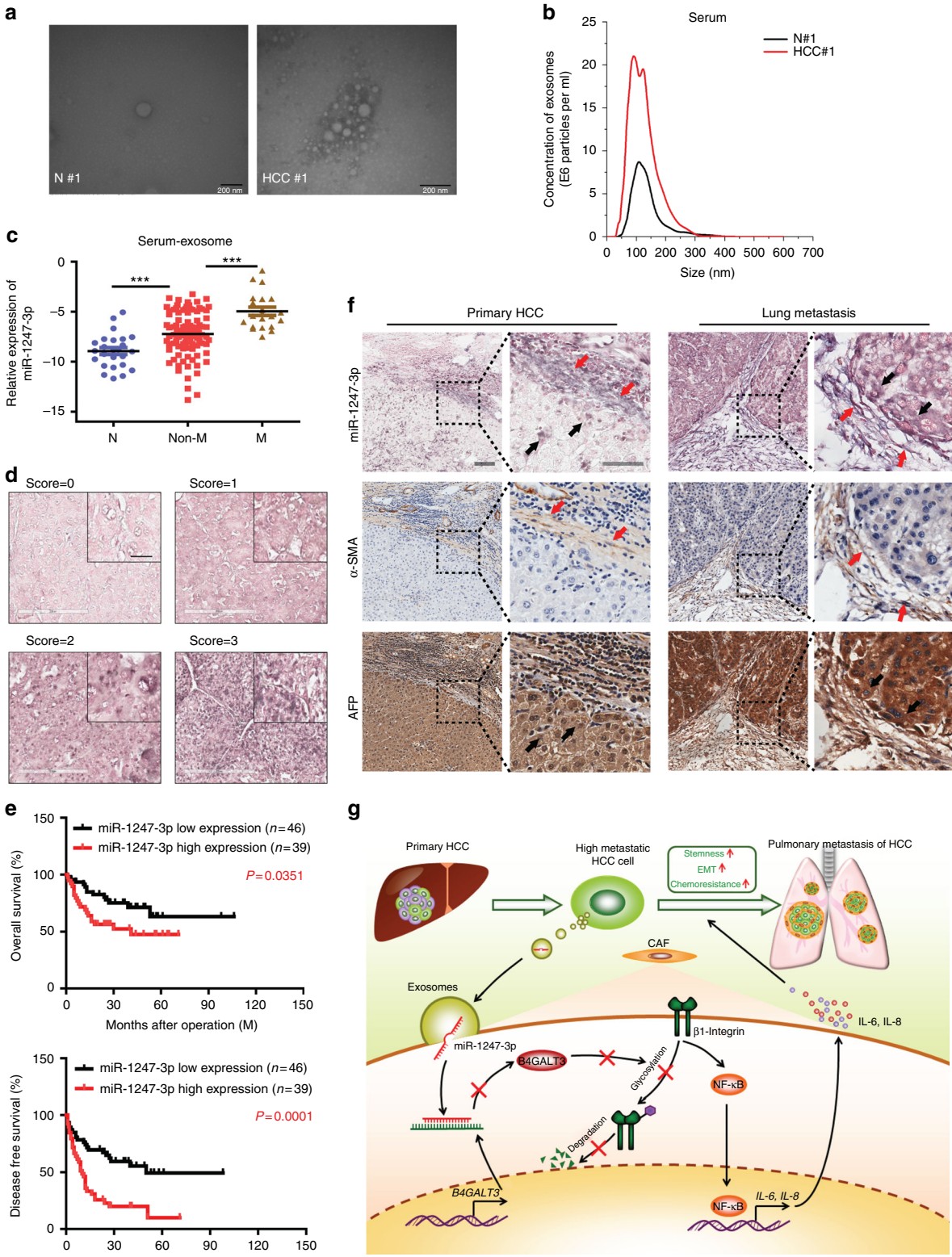

**Fig. 6** MiR-1247-3p is associated with lung metastatic progression in liver cancer. **a**, **b** Exosomes in normal and HCC serums were detected by electron microscopy and Nanosight particle tracking analysis. Scale bar, 200 nm. **c** MiR-1247-3p expression level in serum exosomes from healthy donors and primary HCC patients. Non-M, HCC patients without lung metastasis. M, HCC patients with lung metastasis. Data are presented as mean ± s.d. Student's *t*-test was used to analyze the data. ***$p < 0.001$. **d** In situ hybridization assay of HCC samples and scores of miR-1247-3p level. Representative images were shown. Scale bar, 50 μm. **e** Kaplan–Meier plots of overall survival and disease-free survival of 85 patients with HCC, stratified by expression of miR-1247-3p. Survival data were analyzed by the Kaplan–Meier method and log-rank test. **f** In situ hybridization of miR-1247-3p in combination with IHC staining of HCC markers (AFP) and fibroblast markers (α-SMA) on serial sections of human HCC tissues and lung metastatic tissue samples. Red arrows indicated fibroblasts; black arrows indicated tumor cells. Scale bar, 100 μm. **g** Proposed schematic diagram of tumor exosomal miR-1247-3p-mediating fibroblasts activation to promote lung metastasis of liver cancer

to manufacturer's protocols. Representative fields were photographed and the number of migrated cells per field was counted.

For wound-healing assay, equal number of MRC5 were plated into six-well plates. Then the cell monolayers were wounded with a pipette tip to draw a gap on the plates. After being treated with tumor-derived exosomes, fibroblasts that migrated into the cleared section was observed under microscope at the specific time points.

**RNA extraction and qRT-PCR.** Total RNA was extracted from cells using Trizol reagent (Invitrogen, Carlsbad, CA, USA) according to the manufacturer's instructions. Reverse transcription was performed using Superscript III RT (Invitrogen) and random primers. Real-time PCR was conducted using SYBR Green PCR Master Mix (Applied TaKaRa, Otsu, Shiga, Japan) and performed on ABI PRISM 7300HT Sequence Detection System (Applied Biosystems, Foster City, CA). 18s was used as a reference control. MiRNAs purification and qRT-PCR were performed as described previously[46]. The sequences of all indicated primers were listed in Supplementary Table 2.

**Immunohistochemistry and in situ hybridization analysis.** For immunohistochemistry, the slides were incubated with primary antibodies referred above, which was followed by incubation with horseradish peroxidase-conjugated secondary antibodies (Santa Cruz Biotechnology). Finally, the staining processes were performed with diaminobenzidine colorimetric reagent solution (Dako, Carpinteria, USA) and hematoxylin (Sigma Chemical Co., USA). For in situ hybridization analysis, hsa-miR-1247-3p miRCURY LNA detection probe (Exiqon, Denmark) was used and the total staining processes were carried out according to manufacturer's protocols described before[47]. Images were captured with Aperio ScanScope AT Turbo (Aperio, USA) and assessed with image-scop software (Media Cybernetics, Inc.)

**Isolation and analysis of exosomes.** For exosomes isolation, we first transplanted equal number of different cells into 10 cm plates and changed the culture medium with fresh DMEM-supplemented serum that was depleted of exosomes by centrifuged at $12,000 \times g$ overnight. After 48 h, CM was collected and filtrated through 0.22 μm filters (Millipore, USA). Exosomes in CM or serum samples were isolated by ultracentrifugation according to the standard methods described previously[48]. Ultracentrifugation experiments were performed with Optima MAX-XP (Beckman Coulter, USA). Exosomes were observed by Philips CM120 BioTwin transmission electron microscope (FEI Company, USA) and quantified by NanoSight NS300 (Malvern Instruments Ltd, UK).

**Exosomes tracing.** For exosome-tracing experiments, tumor cells were pre-treated by DiO (Beyotime, China) and exosomes in CM was obtained as described above. After incubation with recipient cells that were pre-treated with DiI (Beyotime), exosomes were observed by confocal laser scanning microscopy TCS SP8 (Leica, Germany).

**Microarray analysis of exosomal miRNAs.** Exosomal miRNAs microarray analysis was performed at Shanghai Biotechnology Corporation (Shanghai, China), using Agilent Human miRNA 8*60 K V21.0 microarray (Agilent Technologies, USA). Quantile normalization and subsequent data processing were performed using Quantile algorithm, Gene Spring Software 12.6 (Agilent Technologies). Hierarchical clustering analysis of the differential expression of miRNAs was performed using the Pearson's correlation analysis with Cluster 3.0 and TreeView software.

**Luciferase reporter assay.** For identifying the binding site between miR-1247-3p and B4GALT3, cells were transfected with a luciferase construct containing B4GALT3 with the wild-type or a mutated version of the binding site, co-transfected with miR-1247-3p mimic or empty vector. For evaluating the luciferase activities of B4GALT3, cells were transfected with a luciferase construct containing B4GALT3 with the wild type or a mutated version of the binding site, pre-treated with tumor-derived exosomes for 24 h. The luciferase vectors were constructed by GenePharma Co. (Shanghai, China). For determining the luciferase activities of NF-κB, cells were transfected with luciferase construct containing NF-κB, pre-treated with tumor-derived exosomes for 24 h. Luciferase activities were detected using the Dual Luciferase kit (Promega, USA) according to the manufacturer's instructions after 48 h of transfection.

**Statistics analysis.** Data analysis was performed using the SPSS software version 16. Each experiment was carried out in triplicate at least and all results were presented as mean ± s.d. $\chi^2$-Test and Student's $t$-test were used to assess statistical significance. Kaplan–Meier analysis and log-rank tests were applied for survival analysis. A value of $p < 0.05$ was considered significant.

**Data availability.** The exosomal miRNA expression data from different liver cancer cells are deposited at Gene Expression Omnibus (accession number: GSE106452). All other remaining data are included in the article and Supplementary Information files, or available from the authors upon reasonable request.

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

## Acknowledgements

We are grateful to Dongping Hu, Dandan Huang, Congli Hu, Huanlin Sun, and Shennian Ge for their technical assistances. This work was supported by National Natural Science Foundation of China (81622039, 81221061, 81572895, 81372356, 81472592, and 81522035).

## Author contributions

W.Y. and H.W. designed the experiments. T.F., H.L., and G.L. performed the experiments, analyzed the data, and wrote the paper. T.L., C.W., Q.H., L.Y., L.G., S.H., D.C., L. T., S.T., and M.W. performed the experiments. B.S. provided the specimens. W.Y. and H. W. organized and supervised the study.

## Additional information

**Competing interests:** The authors declare no competing financial interests.

