## [Peer Review File · Nature Communications]

Reviewers' comments:

Reviewer #1 (Remarks to the Author):

In this manuscript, Fang et al. describe that high-metastatic hepatocellular carcinoma (HCC) cell derived extracellular vesicles (EVs) regulated the activation of lung fibroblasts to create the metastatic niche. They also showed that miR-1247-3p were enriched in high-metastatic HCC cell derived EVs compared with non-metastatic HCC derived EVs and targeted B4GALT3 to activate the β 1-integrin-NF- κ B signaling in lung fibroblast cell line. The concept and content of this manuscript are certainly interesting. However, several points should be addressed especially in the link between fibroblasts education by EVs and HCC progression.

Specific comments:

1. In figure 1 C, the authors showed the expression of HSP70, CD63 and TSG101 in exosome. However, it is not sufficient. The authors should investigate at least more two exosome markers including CD9, CD81.
2. In page 5, line 15 to line 17, the authors mentioned about α -SMA positive CAF within lung metastatic niche. However, in this manuscript, the authors only focused on the relationship between HCC cell-derived exosome and inflammatory cytokines expression of CAF. If highly-metastatic HCC cell-derived exosome contribute to the reprogramming of CAF in lung metastatic niche, it is also conceivable that exosome also affect the several kinds of CAF phenotypes including cell contraction and the expression of α -SMA, CXCL12, TGF- β and collagen (Orimo A et al., *Cell*, 121(3):335-48, 2005) (Kalluri R et al., *Nat. Rev. Cancer*, 6(5): 392-401, 2006). The authors should investigate whether highly-metastatic HCC cell-derived exosome affect the expression of CAF markers and cell contraction ability.
3. The authors mentioned that miR-1247-3p was identified by microarray profiling of exosomes. However, it is still unclear the expression level of miR-1248-3p in HCC cells, MRC5 cells and HCC cell-derived exosomes. Microarray expression data is insufficient to prove that miR-1247-3p was really contained in exosomes and HCC cell-derived exosomes played important role in cancer microenvironment. The authors should quantify miR-1247-3p expression in HCC cell lines, MRC5 and HCC-derived exosomes by qRT-PCR or next generation sequencing.
4. The localization of miR-1247-3p expression in HCC tissue and lung metastatic tissue sample is not clear. The author should also perform in situ hybridization of miR-1247-3p in combination with immunofluorescence staining of HCC markers and fibroblast markers in HCC tissue and lung metastatic tissue sample.
5. In figure 2E, the authors showed the migration ability of MRC5 with exosomes in presence of miR-1247-3p inhibitor or not. The authors should also investigate the migration ability of MRC5 with exosome that are derived from highly metastatic HCC cells with stably expressing miR-1247-3p inhibitor.
6. Although the author showed the relationship between miR-1247-3p expression and B4GALT3, they did not show the effect of highly-metastatic HCC cells derived exosomes on the expression of B4GALT3. If miR-1247-3p expression is enriched in exosomes, treatment of highly-metastatic HCC cells derived exosomes also suppress B4GALT3 expression in MRC5 cell lines. The authors should investigate the effect of exosomes on B4GALT3 expression in MRC5 by western blot and 3'-UTR analysis.
7. In figure 5, the authors showed the effect of miR-1247-3p transfected fibroblasts on HCC progression. However, these data are unsatisfied to prove that HCC cell-derived exosome reprogram CAF phenotype to promote HCC progression. The authors should investigate whether fibroblasts educated by highly-metastatic HCC cell-derived exosomes promote the expression of stemness genes, spheroid formation ability, motility, EMT process and resistance ability to sorafenib.
8. The relationship between exosome-induced inflammatory cytokines and cancer progression is

obscure. The authors should investigate whether the effects of fibroblasts educated by miR-1247-3p and exosome on the HCC progression are abolished by inhibition of inflammatory cytokine expression.

Minor comments:

In figure 1A and 6A, the authors showed the image of exosome by electron microscopy. However, these data are not clear. The authors should show more clear images.

Reviewer #2 (Remarks to the Author):

In this manuscript, the authors found that high-metastatic HCC cells had stronger ability in reprogramming normal fibroblasts to CAFs than low-metastatic HCC cells. HCC secreted exosomal miR-1247-3p might promote CAF reprogram by targeting B4GALT3 and subsequently activated β 1-integrin-NF- κ B signaling. Clinical data found that high serum exosomal miR-1247-3p was correlated with lung metastasis in HCC. This is an interesting work and follows are few comments:

1. Are there more CAF specific markers to distinguish CAF from normal fibroblasts besides α -SMA?
2. What is the difference between "reprogramming normal fibroblasts to CAFs" and "fibroblasts activation"? To me it looks the same thing described in two ways. However, cell reprogram leads to dedifferentiation, while cell activation results in maturation.
3. When MRC5 cells was treated with miR-1247-3p, IL-1 β , IL-6 and IL-8 were up-regulated. Authors should define whether this effect is a direct regulatory result or is caused by the activation of fibroblasts. The simple way is to test if these inflammatory genes could be up-regulated in other cells after miR-1247-3p treatment?

Responses to Comments by Reviewer 1:

In this manuscript, Fang et al. describe that high-metastatic hepatocellular carcinoma (HCC) cell derived extracellular vesicles (EVs) regulated the activation of lung fibroblasts to create the metastatic niche. They also showed that miR-1247-3p were enriched in high-metastatic HCC cell derived EVs compared with non-metastatic HCC derived EVs and targeted B4GALT3 to activate the β 1-integrin-NF- κ B signaling in lung fibroblast cell line. The concept and content of this manuscript are certainly interesting. However, several points should be addressed especially in the link between fibroblasts education by EVs and HCC progression.

Specific comments:

1. In figure 1 C, the authors showed the expression of HSP70, CD63 and TSG101 in exosome. However, it is not sufficient. The authors should investigate at least more two exosome markers including CD9, CD81.

Thanks to the reviewer's constructive comment. Based on this suggestion, additional experiments have been performed. As shown in revised Fig 1C, CD9 and CD81 were also detected in the isolated exosomes. This issue has been addressed on page 5 in the revised manuscript.

2. In page 5, line 15 to line 17, the authors mentioned about α -SMA positive CAF within lung metastatic niche. However, in this manuscript, the authors only focused on the relationship between HCC cell-derived exosome and inflammatory cytokines expression of CAF. If highly-metastatic HCC cell-derived exosome contribute to the reprogramming of CAF in lung metastatic niche, it is also conceivable that exosome also affect the several kinds of CAF phenotypes including cell contraction and the expression of α -SMA, CXCL12, TGF- β and collagen (Orimo A et al., Cell, 121(3):335-48, 2005) (Kalluri R et al., Nat. Rev. Cancer, 6(5): 392-401, 2006). The authors should investigate whether highly-metastatic HCC cell-derived exosome affect the expression of CAF markers and cell contraction ability.

We thank the reviewer for this constructive comment. We further examined the expression of several CAF markers (α -SMA, CXCL12, TGF- β , COL1A1, COL3A1, COL4A1) and the ability of fibroblasts to contract collagen gel. As shown in Fig 1F and 1G., expression of CAF markers and the ability to contract collagen gel were increased in the MRC5 cells treated with highly-metastatic HCC cell-derived exosomes. These issues have been addressed on page 6 in the revised manuscript.

3. The authors mentioned that miR-1247-3p was identified by microarray profiling of exosomes. However, it is still unclear the expression level of miR-1247-3p in HCC cells, MRC5 cells and HCC cell-derived exosomes. Microarray expression data is insufficient to prove that miR-1247-3p was really contained in exosomes and HCC cell-derived exosomes played important role in cancer microenvironment. The authors should quantify miR-1247-3p expression in HCC cell lines, MRC5 and HCC-derived exosomes by qRT-PCR or next generation sequencing.

We detected miR-1247-3p expression by qRT-PCR in HCC cell lines, MRC5, and HCC-derived exosomes. As shown in Fig S2A, miR-1247-3p was expressed at different levels in HCC cell lines, MRC5 and HCC-derived exosomes. Furthermore, the expression of miR-1247-3p was elevated in both two high-metastatic cancer cells and high-metastatic cancer cells-derived exosomes. These issues have been addressed on page 7 in the revised manuscript.

4. The localization of miR-1247-3p expression in HCC tissue and lung metastatic tissue sample is not clear. The author should also perform in situ hybridization of miR-1247-3p in combination with immunofluorescence staining of HCC markers and fibroblast markers in HCC tissue and lung metastatic tissue sample.

Thanks to the reviewer's constructive comment. In situ hybridization of miR-1247-3p and IHC staining using anti-SMA antibody and anti-AFP antibody were performed on serial sections of human HCC tissues and lung metastatic tissues. As shown in revised Fig.6F, miR-1247-3p signals were present in both tumor cells and some fibroblasts around the tumor. These issues have been addressed on page 11 in the revised manuscript.

5. In figure 2E, the authors showed the migration ability of MRC5 with exosomes in presence of miR-1247-3p inhibitor or not. The authors should also investigate the migration ability of MRC5 with exosome that are derived from highly metastatic HCC cells with stably expressing miR-1247-3p inhibitor.

Per the reviewer's suggestion, a lentiviral delivery system was applied to stably express miR-1247-3p inhibitor or negative control in highly metastatic HCC cells (CSQT-2 and HCC-LM3). The knock-down effects were verified in both HCC cells and exosomes derived (Fig S2D). As shown in Fig 2E and Fig S2E, the increased migration ability of MRC5 educated with highly metastatic HCC cells-derived exosomes was inhibited by stable miR-1247-3p inhibitor. These issues have been addressed on page 7 in the revised manuscript.

6. Although the author showed the relationship between miR-1247-3p expression and B4GALT3, they did not show the effect of highly-metastatic HCC cells derived exosomes on the expression of B4GALT3. If miR-1247-3p expression is enriched in exosomes, treatment of highly-metastatic HCC cells derived exosomes also suppress B4GALT3 expression in MRC5 cell lines. The authors should investigate the effect of exosomes on B4GALT3 expression in MRC5 by western blot and 3'-UTR analysis.

Thanks to the reviewer's constructive comment. MRC5 was treated with exosomes derived from HCC cells with different metastatic potential, and then B4GALT3 expression levels in MRC5 were detected by western blot. As shown in Fig S3C-3D, treatment with highly-metastatic HCC cells derived exosomes suppressed B4GALT3 expression in MRC5 cells. Moreover, relative luciferase activity of B4GALT3 in MRC5 treated with different tumor exosomes was also tested. As shown in Fig S3E-3F, the luciferase activity significantly decreased only in MRC5 transfected with wild-type binding site vector of B4GALT3 after treatment with high-metastatic cancer cells-derived exosomes. These issues have been addressed on page 8 in the revised manuscript.

7. In figure 5, the authors showed the effect of miR-1247-3p transfected fibroblasts on HCC progression. However, these data are unsatisfied to prove that HCC cell-derived exosome reprogram CAF phenotype to promote HCC progression. The authors should investigate whether fibroblasts educated by highly-metastatic HCC cell-derived exosomes promote the expression of stemness genes, spheroid formation ability, motility, EMT process and resistance ability to sorafenib.

Per the reviewer's suggestion, we conducted additional experiments to examine whether MRC5 educated by highly-metastatic HCC cell-derived exosomes can also promote HCC progression.

As shown in Fig S5A, exosomes derived from high-metastatic cancer cells enhanced the secretion of IL-6 and IL-8 in fibroblasts. As shown in Fig S5B-S5G, conditional medium collected from MRC5 cells pre-treated by highly-metastatic HCC cell-derived exosomes promoted the expression of stemness genes, spheroid formation ability, motility, EMT process and resistance ability to sorafenib of SMMC7721 cells. These issues have been addressed on page 9-10 in the revised manuscript.

8. The relationship between exosome-induced inflammatory cytokines and cancer progression is obscure. The authors should investigate whether the effects of fibroblasts educated by miR-1247-3p and exosome on the HCC progression are abolished by inhibition of inflammatory cytokine expression.

We thank the reviewer for raising this important issue. To investigate the role of inflammatory cytokine in regulating the tumor-promoting functions of fibroblasts, we used anti-IL-6 or anti-IL-8 antibodies to block IL-6 signaling or IL-8 signaling, respectively. As shown in Fig 5D, 5G, 5I and Fig S5E-5G, IL-6 or IL-8 antibody neutralization, but not IgG, resulted in a significant decrease in the tumor-promoting functions of MRC5 educated by miR-1247-3p or exosomes. These data suggested that inflammatory cytokines played important roles in the tumor-promoting functions of fibroblasts. These issues have been addressed on page 9-10 in the revised manuscript.

Minor comments:

In figure 1A and 6A, the authors showed the image of exosome by electron microscopy. However, these data are not clear. The authors should show more clear images.

We have re-conducted this experiment and improved electron microscopy images were shown in Fig 1A and Fig 6A in the revised manuscript.

Responses to Comments by Reviewer 2:

In this manuscript, the authors found that high-metastatic HCC cells had stronger ability in reprogramming normal fibroblasts to CAFs than low-metastatic HCC cells. HCC secreted exosomal miR-1247-3p might promote CAF reprogram by targeting B4GALT3 and subsequently activated β 1-integrin-NF- κ B signaling. Clinical data found that high serum exosomal miR-1247-3p was correlated with lung metastasis in HCC. This is an interesting work and follows are few comments:

1. Are there more CAF specific markers to distinguish CAF from normal fibroblasts besides α -SMA?

Thanks the reviewer for raising this valuable comment. The same concern was also raised by the reviewer #1. We further examined the expression of several CAF markers (α -SMA, CXCL12, TGF- β , COL1A1, COL3A1, COL4A1) and the ability of fibroblasts to contract collagen gel. As shown in Fig 1F-1G, expression of CAF markers and the ability to contract collagen gel, were increased in the MRC5 treated with highly-metastatic HCC cell-derived exosomes. These issues have been addressed on page 6 in the revised manuscript.

2. What is the difference between "reprogramming normal fibroblasts to CAFs" and "fibroblasts activation"? To me it looks the same thing described in two ways. However, cell reprogram leads

to dedifferentiation, while cell activation results in maturation.

Thanks the reviewer for raising this valuable comment. We are so sorry for making such confusion and have replaced “reprogramming normal fibroblasts to CAFs” with “fibroblasts activation”.

3. When MRC5 cells was treated with miR-1247-3p, IL-1 β , IL-6 and IL-8 were up-regulated. Authors should define whether this effect is a direct regulatory result or is caused by the activation of fibroblasts. The simple way is to test if these inflammatory genes could be up-regulated in other cells after miR-1247-3p treatment?

We thank the reviewer for their good suggestions. We treated HCC cells with miR-1247-3p, and detected the expression of these inflammatory genes. As shown in Fig S2B, the expression of IL-1 β , IL-6 and IL-8 was also increased after miR-1247-3p treatment. Thus, these data demonstrated that the increased expression of these inflammatory genes is a direct regulatory result of miR-1247-3p. These issues have been addressed on page 7 in the revised manuscript.

REVIEWERS' COMMENTS:

Reviewer #1 (Remarks to the Author):

Reviewer comment:

The authors addressed all of my concerns. The concept and content of this manuscript are really exciting. Because highly metastatic cancer cell-derived miR-1247 are important for cancer metastasis through educating fibroblasts, it is conceivable that metastatic cells of other cancer cell types (breast cancer, colorectal cancer etc.) might also use same mechanism.

Reviewer #2 (Remarks to the Author):

The responses from authors are satisfactory. I have no more comment.

Responses to Comments by Reviewer #1:

The authors addressed all of my concerns. The concept and content of this manuscript are really exciting. Because highly metastatic cancer cell-derived miR-1247 are important for cancer metastasis through educating fibroblasts, it is conceivable that metastatic cells of other cancer cell types (breast cancer, colorectal cancer etc.) might also use same mechanism.

Thanks to the reviewer's comment.

Responses to Comments by Reviewer #2:

The responses from authors are satisfactory. I have no more comment.

Thanks to the reviewer's comment.